# Epidemiology and clinical profile of diarrhea associated with enterotoxigenic *Escherichia coli* and *Vibrio cholerae* in Bangladesh: Findings from a hospital-based surveillance system, 2008–2023

Jinat Alam[1☉], Shamsun Nahar Shaima[1,2☉], Rina Das[1,3]*, Md. Ahshanul Haque[1], Md. Nasif Hossain[4,5], Soroar Hossain Khan[1], Sharika Nuzhat[1], Mohammod Jobayer Chisti[1], Tahmeed Ahmed[1,6,7,8], Subhra Chakraborty[9‡], A. S. G. Faruque[1‡]

**1** Nutrition Research Division, International Centre for Diarrhoeal Disease Research, Bangladesh (icddr,b), Dhaka, Bangladesh, **2** Department of Nutritional Sciences, School of Graduate Studies, University of Toronto, Toronto, Ontario, Canada, **3** Gangarosa Department of Environmental Health, Rollins School of Public Health, Emory University, Atlanta, Georgia, United States of America, **4** University of Virginia Environmental Institute, University of Virginia, Charlottesville, Virginia, United States of America, **5** Department of Public Health Sciences, University of Virginia School of Medicine, Charlottesville, Virginia, United States of America, **6** Office of the Executive Director, International Centre for Diarrhoeal Disease Research, Bangladesh (icddr,b), Dhaka, Bangladesh, **7** Department of Global Health, University of Washington, Seattle, Washington, United States of America, **8** Department of Public Health Nutrition, James P Grant School of Public Health, BRAC University, Dhaka, Bangladesh, **9** Department of International Health, Johns Hopkins Bloomberg School of Public Health, JHU, Baltimore, Maryland, United States of America

☉ These authors contributed equally to this work.
‡ SC and ASGF also contributed equally as senior author to this work.
\* rina.das@icddrb.org, rina.das@emory.edu

## Abstract

### Background

Enterotoxigenic *Escherichia coli* (ETEC) and *Vibrio cholerae* are notable enteric bacterial pathogens that cause diarrheal illnesses in resource-limited settings (e.g., Bangladesh). We aimed to explore the epidemiology and the clinical presentation of diarrhea caused by either *V. cholerae* or ETEC as a single-pathogen, or as a co-infection with both pathogens among patients requiring hospitalization.

### Methods

We conducted the study using data from the Diarrheal Diseases Surveillance System of Dhaka Hospital, icddr,b from 2008 to 2023. A multivariable logistic regression model was used to characterize association of identified bacterial pathogens with specific clinical features.

### Results

Among 43,483 diarrheal patients who received hospital care, 11% (4540/43,483) tested positive for *V. cholerae,* 8% (3070/43,483) had ETEC, and 1.5% (630/43,483)

**Data availability statement:** Based on the recommendation of its ethical review board, the research administration of the International Centre for Diarrhoeal Disease Research, Bangladesh (icddr,b) has restricted making the personal information of hospitalized patients publicly available. The dataset analyzed in this study is part of a larger dataset containing personal information of patients seeking care from hospital, including names, admission dates, and residential areas. Making such data publicly available could compromise patient privacy and contravene the policy of Diarrheal Disease Surveillance System approved by our institutional review boards. To protect patient confidentiality, icddr,b's policy prohibits sharing the full dataset in publications, supplemental files, or public repositories. However, a portion of the dataset relevant to this study is available upon request for researchers meeting access criteria. Interested researchers may contact Ms. Shiblee Sayeed, Senior Manager, Research Administration, at shiblee_s@icddrb.org, at icddr,b for data access inquiries.

**Funding:** The author(s) received no specific funding for this work.

**Competing interests:** The authors have declared that no competing interests exist.

had co-infection. In 2023, the frequencies of ETEC, *V. cholerae,* and co-infection with both pathogens among patients were 7.5%, 4.5%, and 7.5%, respectively. After adjusting for covariates, co-infected cases showed significantly higher odds of severe outcomes, including watery stools (aOR: 12.33), high stooling frequency (>10/day, aOR: 1.50), vomiting (aOR: 3.16), and intravenous rehydration (aOR: 8.70) compared to single-pathogen infections. Clinical features among patients infected with single pathogens also varied. *V. cholerae*-positive cases were associated with dehydration [aOR:5.64;95%CI:(4.94,6.43)] and length of hospital stay [aOR:1.81;95%CI:(1.68,1.94)] relative to *V. cholerae*-negative cases. ETEC-positive cases were more likely to present with watery stools [aOR:1.26;95%CI:(1.04,1.53)], dehydration [aOR:1.23;95%CI:(1.12,1.35)], and the requirement for intravenous fluid rehydration [aOR:1.15;95%CI:(1.04,1.27)] relative to ETEC-negative cases. Overall, the clinical presentations of patients with ETEC single infection were less severe compared to patients with *V. cholerae* as a single infection or co-infection.

## Conclusions

Co-infection with *V. cholerae* and ETEC results in more severe clinical manifestations requiring intensive medical management compared to single-pathogen infections. These findings highlight the need for enhanced clinical preparedness and consideration of testing for both pathogens to optimize patient care. Our findings highlight the potential value of vaccines targeting ETEC and *V. cholerae* to improve protection.

## Author summary

Diarrheal diseases continue to pose significant public health challenges, particularly in low- and middle-income countries (LMICs). Approximately 1.5 million deaths worldwide are attributed to diarrheal diseases. Enterotoxigenic *Escherichia coli* (ETEC) and *Vibrio cholerae* are notably frequent enteric bacterial pathogens that cause diarrheal illnesses in resource-limited settings (e.g., Bangladesh). This study was undertaken using a robust dataset from the largest diarrheal disease hospital, spanning 16 years. This approach allowed us to analyze trends, patterns, and changes over time. This study describes the epidemiology and the clinical presentation of diarrhea caused by either *V. cholerae* or ETEC as a single-pathogen, or as a co-infection with both pathogens, among patients requiring hospitalization. This research focused on a particular location in South Asia, urban Bangladesh. But ETEC and *Vibrio cholerae* are global pathogens. This study's findings are relevant for appreciating the problems faced by people who experience illness caused by these pathogens, particularly those who live in cities in LIMCs. These findings highlight the need for enhanced clinical preparedness and consideration of testing for both pathogens to optimize patient care. Our findings also highlight the potential value of vaccines targeting ETEC and *V. cholerae* to improve protection.

## Introduction

Diarrheal diseases are a significant public health challenge, with greater prevalence in low- and middle-income countries (LMICs) [1]. Approximately 1.5 million deaths worldwide are attributed to diarrheal diseases [1,2]. It is regarded as the third leading cause of death for children under five and accounts for 443,832 child deaths annually [1]. Worldwide, acute watery diarrheal outbreaks, especially cholera epidemics, are serious health concerns [3]. Globally, the burden of cholera is greatest in LMICs, where 1.4 billion people are at risk for cholera in endemic countries [4]. The World Health Organization (WHO) estimates that cholera affects 3–5 million people annually, resulting in up to 120,000 deaths [5]. Despite WHO recommendations for vaccines and other preventive measures, case-fatality rates remain higher than expected, underscoring the need for improved control strategies [6]. In Dhaka, the capital city of Bangladesh, a bimodal cholera epidemic is an annual phenomenon [7].

According to WHO estimates, enterotoxigenic *Escherichia coli* (ETEC) accounts for an estimated 280–400 million cases of diarrhea annually, with 18,700–42,000 deaths in children under five years of age [8]. While the disease burden of ETEC is primarily documented in young children, data on older children and adults remain sparse [9]. These two enteric bacterial pathogens *Vibrio cholerae* and ETEC are among the more frequently identified causes of watery diarrhea among hospitalized patients in Bangladesh [10]. In Bangladesh, co-infections with ETEC and *V. cholerae* have also been reported, with 12–14% of diarrheal patients presenting with co-infections, often associated with severe dehydration [11]. Both pathogens share similar modes of transmission, exhibiting overlapping seasonal peaks during spring (March–May) and post-monsoon (September–November) seasons [12,13].

Epidemiological studies often overlook the impact of co-infections, leading to an incomplete picture of diarrheal disease burden across different populations [14]. To close this knowledge gap we aimed to examine the clinical presentation and disease severity among patients with diarrhea caused by *V. cholerae,* ETEC, or co-infection with both pathogens. We analyzed the demography of the patients, and the risk factors associated with these three groups of patients.

## Methods

### Ethics statement

The studies involving human participants were reviewed and approved by Research Review Committee (RRC) and Ethical Review Committee (ERC) of International Centre for Diarrhoeal Disease Research, Bangladesh (icddr,b). Verbal consent was taken from each of the patients or from the guardians or caregivers of the patients of 2% systematic surveillance of 'DDSS' following hospitalization, assuring them of the confidentiality of information, and taking permission to use the data aimed at analysis for the results. Verbal consent was also obtained from the parents or legal guardians of all child participants involved in the 2% systematic surveillance of 'DDSS'. All stored information is used for conducting this research and publications.

### Ethical considerations

The 2% systematic surveillance of 'DDSS' at icddr,b Dhaka Hospital is a routine ongoing process. This surveillance used an anonymized protocol for patient records and obtained informed consent from the participants or their parents, or legal guardians before enrolling them. All research was performed following the relevant guidelines and regulations.

### Study setting

icddr,b (International Centre for Diarrhoeal Disease Research, Bangladesh) is a public health research institute, located in Dhaka, Bangladesh. In addition to research, every year, icddr,b provides clinical care to diarrheal patients with over 250,000 cases managed in its treatment facilities [15].

## Diarrheal Disease Surveillance System (DDSS)

The Diarrheal Disease Surveillance System (DDSS) was established at the Dhaka Hospital in 1979 [16]. The DDSS includes a 2% sample of patients of all age groups visiting Dhaka Hospital [16]. Trained health research staff interview every 50th patient or their attendants using structured questions to gather information on socioeconomic and demographic details. Information on housing and environmental conditions, child feeding practices, and use of medications and fluids at home is collected. They also record clinical symptoms, anthropometry, treatments received at the facility, and patient outcomes. Microbiological assessments of fecal samples (culture, multiplex polymerase chain reaction and enzyme-linked immunosorbent assay) are performed to identify common diarrheal pathogens such as *V. cholerae*, ETEC, *Shigella*, *Salmonella*, *Campylobacter,* and Rotavirus.

## Study design, study population, sampling technique, and eligibility criteria

The current study used information collected from participants of both sexes and all age groups included in the DDSS database of the Dhaka Hospital of icddr,b. The analysis in this study was conducted with data collected over sixteen years, between January 2008 to December 2023. The 2008–2023 period was chosen based on our objectives, as it marks the earliest availability of ETEC data. **Fig 1** illustrates the sampling strategy from total diarrheal patients and sample selection procedure.

## Operational definitions

**Diarrhea.** Diarrhea is defined as the passage of three or more loose or liquid stools per day [17].

**Household asset index.** The asset index is a composite measure of a household's cumulative living standard. It was calculated using data on a household's ownership of selected assets, such as televisions and bicycles, materials used for housing construction, as well as access to clean water and sanitation facilities [18]. Principal components analysis was used to create an asset index, which categorizes individuals as poor, lower middle, middle, upper middle, or rich [19–21].

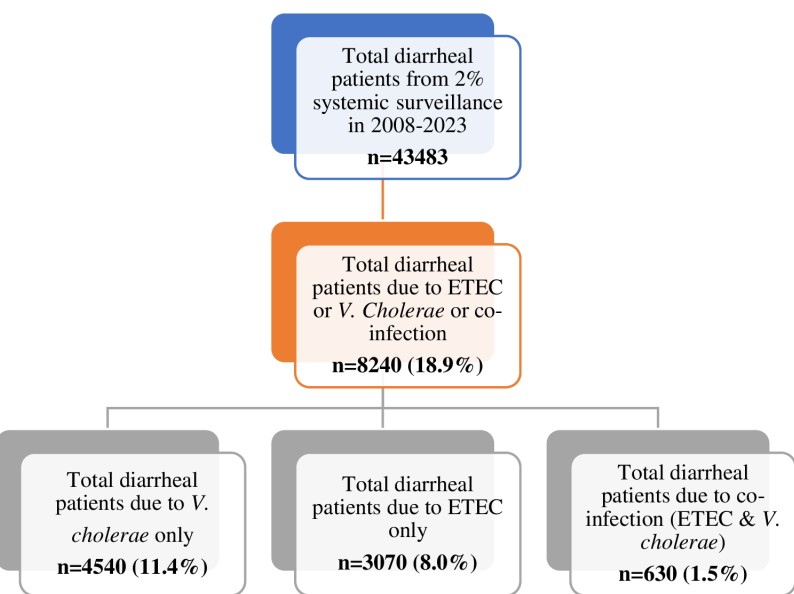

**Fig 1. Sampling strategy from total diarrheal patients and sample selection procedure.**

## Outcome variables and covariates

The outcome variables of this study were the clinical characteristics of the participants including; duration of diarrhea before arrival, character of stool (non-watery or watery), presence of blood in stool, number of stools in last 24 hours, vomiting in last 24 hours, abdominal pain, fever (temperature ≥ 38°C), dehydration status, rehydration method used, and length of hospitalization. Infection due to *V. cholerae* or ETEC or co-infection with both *V. cholerae* and ETEC were the main exposure variables. Covariates were the socio-demographic characteristics: age, sex, area of residence, household asset index, use of rehydration solution [Oral Rehydration Solution (ORS) packet/homemade ORS/barley/rice gruel/IV fluid/rice saline] before arrival, and use of antibiotics before hospitalization.

## Statistical analysis

Descriptive statistics for all the demographic and clinical characteristics data were presented using frequency with percentages (%). Frequency (%) of ETEC, *V. cholerae* and co-infection patients was calculated from hospital records data. A multivariable logistic regression model was employed to assess the relationship between the pathogen of interest and the clinical presentation of diarrheal cases. Each separate model for the variables (clinical features) was adjusted for age group, sex, area of residence, mother's education, father's education, household asset index, use of rehydration solution, and antibiotics before hospitalization. The analysis was done separately for ETEC positive cases (ETEC negative cases = reference), *V. cholerae* positive cases (*V. cholerae* negative cases = reference), and also for co-infection with both ETEC and *V. cholerae* (*n*o ETEC or *V. cholerae* = reference*)*. A total of thirty models (individual logistic regression) were performed for the final analysis. P-values <0.05 (α) were considered significant for a single hypothesis test. However, we needed to control for the family wise alpha inflation to interpret logistic regression findings since we analyzed ten outcomes that were not independent. 'Sidak correction' was used to control the family wise error rate [22]. The new level of significance was calculated, $\alpha_{new} = [1 - (1 - \alpha_{old})^{1/n}] = [1 - (1 - 0.05)^{1/10}] = 0.005$. Statistical software, "Stata SE 17.0 version" and R Statistical software (version 4.2.3) were used for data analysis. Using Microsoft Excel 2019, graphs and plots were produced for visualization.

## Role of the funding source

The author(s) received no dedicated funding or financial support for the research, authorship, and/or publication of this article.

## Results

From 2008 to 2023, 43,483 patients with diarrhea were surveilled. Of these, 8240 were found to have a stool culture positive for ETEC, *V. cholerae,* or both pathogens.

## Demographic characteristics of patients with *V. cholerae*, ETEC, and co-infection, 2008–2023

Among a total of 43,483 patients, *V. cholerae* in isolation was cultured in 11% (n = 4540) of patients. Among patients with *V. cholerae* alone, 59% were male. Just 7% were from a slum area. Sixty-seven percent of mothers had no formal schooling, 66% used tap water, and only 66% drank treated (boiled) water. Sanitary or semi-sanitary toilets were used by 85%, and 73% used ORS. Antibiotics were used as well as 61% prior to reporting to the facility. Overall, 8% of patients (n = 3070) were affected by ETEC. Among these, 6% were from slums, 67% used tap water, and 58% drank treated water (boiled/used filter/used alum/tablet/sieving). Approximately 75% used ORS, and 63% had antibiotics at home. Nearly 1.5% of surveyed cases presented with co-infection (n = 630) with both *V. cholerae* and ETEC cultured from stool. Fifty-seven percent of these co-infected patients were male, 7% were from slum areas, 71% of the mothers had no formal schooling, 70% used tap water, and 62% drank treated water. Overall, 79% of all respondents with co-infection used ORS and 59%

took antibiotics at home. The majority of families belonged to the lower middle-income group in these cases of co-infection (Table 1).

Fig 2 illustrates the age-specific distribution of *V. cholerae*, ETEC, and co-infection. Among identified cholera cases, the 15–30 years age group bore the greatest burden. In contrast, ETEC infections were more frequent (27.5%) among

**Table 1. Baseline demographic characteristics of patients with V. cholerae, ETEC, and co-infection, 2008-2023.**

| Variables | Cholera (+) n = 4540 (11.4%) | ETEC (+) n = 3070 (8.0%) | Co-infection with both Cholera and ETEC n = 630 (1.5%) | No Cholera or ETEC infection n = 35243 (79.1%) |
|---|---|---|---|---|
| **Sex** | | | | |
| Male | 2681 (59.1) | 1763 (57.5) | 356 (56.5) | 20580 (58.7) |
| Female | 1858 (40.9) | 1304 (42.5) | 274 (43.5) | 14460 (41.3) |
| **Area of residence** | | | | |
| Slum area | 344 (7.6) | 193 (6.3) | 44 (7.0) | 1446 (4.2) |
| Other areas* | 4195 (92.4) | 2871 (93.7) | 586 (93.0) | 32744 (95.8) |
| **Years of schooling (mother)** | | | | |
| One or more years of schooling | 1489 (32.8) | 1608 (52.5) | 182 (28.9) | 22178 (64.9) |
| No schooling | 3049 (67.2) | 1454 (47.5) | 448 (71.1) | 11994 (35.1) |
| **Years of schooling (father)** | | | | |
| One or more years of schooling | 1649 (36.3) | 1644 (53.7) | 218 (34.6) | 21987 (64.3) |
| No schooling | 2889 (63.7) | 1418 (46.3) | 412 (65.4) | 12189 (35.7) |
| **Source of drinking water** | | | | |
| Tap water | 3013 (66.4) | 2061 (67.1) | 440 (69.8) | 20376 (57.8) |
| Tube well water | 1527 (33.6) | 1009 (32.9) | 190 (30.2) | 14867 (42.2) |
| **Type of drinking water** | | | | |
| Treated** | 3021 (66.5) | 1789 (58.3) | 389 (61.7) | 23979 (68.0) |
| Not treated | 1519 (33.5) | 1281 (41.7) | 241 (38.2) | 11264 (32.0) |
| **Type of toilet used** | | | | |
| Sanitary/semi-sanitary | 3864 (85.1) | 2697 (87.8) | 565 (89.7%) | 29585 (83.9) |
| Non-sanitary | 676 (14.9) | 373 (12.2%) | 65 (10.3%) | 5658 (16.1) |
| **Use of oral rehydration solution before hospital arrival** | | | | |
| Used | 3353 (73.9) | 2319 (75.5) | 500 (79.4) | 23797 (67.5) |
| Not used | 1187 (26.1) | 751 (24.5) | 130 (20.6) | 11446 (32.5) |
| **Antibiotics taken before hospitalization** | | | | |
| Used | 2782 (61.3) | 1931 (63.0) | 370 (58.7) | 25662 (75.1) |
| Not used | 1757 (38.7) | 1134 (37.0) | 260 (41.3) | 8528 (24.9) |
| **Household Asset Index** | | | | |
| Poor | 890 (19.6) | 459 (14.9) | 105 (16.7) | 5936 (16.8) |
| Lower middle | 1341 (29.5) | 636 (20.7) | 185 (29.4) | 7205 (20.4) |
| Middle | 814 (17.9) | 568 (18.5) | 96 (15.2) | 7003 (19.9) |
| Upper middle | 792 (17.4) | 691 (22.5) | 122 (19.4) | 8375 (23.8) |
| Rich | 703 (15.5) | 716 (23.3) | 122 (19.4) | 6724 (19.1) |

* Other areas of residence included a common housing area, a residential area, and a village area.

** Water was treated through methods such as boiling, filtration, using alum/chlorine tablets, or sieving.

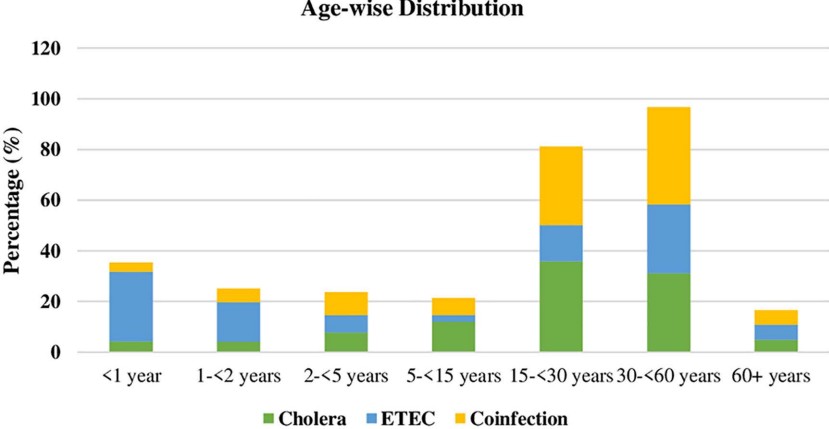

**Fig 2. Changes over the years in *V. cholerae*, ETEC, and co-infection.**

patients aged <1 year. Co-infection cases were clustered among adults aged 30–60 years, who made up 38.4% of all such cases.

### Clinical features of patients with *V. cholerae*, ETEC, and co-infection, 2008–2023

Clinical features varied significantly among *V. cholerae*, ETEC, and co-infection cases. *V. cholerae* infections were notably more severe, with 93% experiencing dehydration and 80% requiring intravenous (IV) rehydration, compared to 62% and 32%, respectively, for ETEC cases. Co-infections closely mirrored *V. cholerae* severity, with 92% presenting with dehydration and 81% needing intravenous rehydration. Vomiting was more common in *V. cholerae* (90%) and co-infection (91%) cases, relative to ETEC cases (73%) (**Table 2**).

Age-related severity patterns differed distinctly among pathogens. *V. cholerae* infections tended to be more severe in patients older than 60 years, while ETEC showed peak severity in children under 5 years. Notably, co-infections exhibited increased severity in both the 5–15 years and over 60 years age groups. This bimodal age-related distribution of acuity suggests a complex interplay between pathogens and host factors across different life stages (**S1 Table**).

### Association of clinical outcomes of the patients with diarrhea caused by *V. cholerae* or ETEC, or co-infection

The multivariable logistic regression analysis revealed significant associations of each clinical feature (outcomes) and pathogens (exposures). After adjusting for covariates, *V. cholerae*-positive cases were associated with an increased likelihood of specific clinical characteristics compared to *V. cholerae*-negative cases. These characteristics included watery stools, more than 10 stools per day, vomiting, dehydration, the need for intravenous rehydration, and hospital stays lasting 24 hours or longer. ETEC positive cases were more likely to have watery stools, dehydration, a requirement for IV fluid rehydration, relative to ETEC negative cases (**Fig 3** and **S2 Table**). Co-infected cases were more likely to have duration of diarrhea < 1 day prior to hospitalization, watery stools, more than 10 stools per day, vomiting, dehydration requiring intravenous rehydration, and hospitalizations ≥ 24 hours (**Fig 3** and **S2 Table**).

### Changing trends by frequency among hospitalized patients (%)

Frequency among hospitalized patients with diarrhea yielded a potential trend into ETEC, *V. cholerae* and co-infection cases from 2008 to 2023. Detection of co-infections increased starting in 2017, reaching 7.5% in 2023. The frequency of

**Table 2. Clinical features of patients with V. cholerae, ETEC, and co-infection, 2008-2023.**

| Variables | Cholera (+) n=4540 (11.4%) | ETEC (+) n=3070 (8.0%) | Co-infection with both Cholera and ETEC n=630 (1.5%) | No Cholera or ETEC infection n=35243 (79.1%) |
|---|---|---|---|---|
| **Duration of diarrhea before arrival at hospital** | | | | |
| <1 day | 3124 (68.8) | 1559 (50.8) | 467 (74.1) | 13771 (39.1) |
| >1day | 1416 (31.2) | 1511 (49.2) | 163 (25.9) | 21472 (60.9) |
| **Character of stool** | | | | |
| Non-watery | 42 (0.9) | 118 (3.8) | 2 (0.3) | 1775 (5.0) |
| Watery | 4498 (99.1) | 2952 (96.2) | 628 (99.7) | 33468 (95.0) |
| **Presence of blood in stool** | | | | |
| Absent | 4276 (94.2) | 2558 (83.3) | 596 (94.6) | 27123 (77.0) |
| Present | 264 (5.8) | 512 (16.7) | 34 (5.4) | 8120 (23.0) |
| **Number of stools in last 24 hours** | | | | |
| ≤10 times | 1307 (28.8) | 1294 (42.2) | 164 (26.0) | 12875 (36.5) |
| >10 times | 3233 (71.2) | 1776 (57.8) | 466 (74.0) | 22368 (63.5) |
| **Vomiting in the last 24 hours** | | | | |
| No | 471 (10.4) | 822 (26.8) | 59 (9.4) | 9000 (25.5) |
| Yes | 4069 (89.6) | 2248 (73.2) | 571 (90.6) | 26243 (74.5) |
| **Abdominal pain** | | | | |
| No | 2251 (49.6) | 1487 (48.4) | 319 (50.6) | 16684 (47.3) |
| Yes | 2289 (50.4) | 1583 (51.6) | 311 (49.4) | 18559 (52.7) |
| **Fever (Temperature >38°C)** | | | | |
| Absent | 4462 (98.3) | 2957 (96.3) | 614 (97.5) | 32811 (93.1) |
| Present | 78 (1.7) | 113 (3.7) | 16 (2.5) | 2432 (6.9) |
| **Assessment of dehydration** | | | | |
| No dehydration | 303 (6.7) | 1157 (37.7) | 47 (7.5) | 16302 (46.3) |
| Dehydration | 4237 (93.3) | 1913 (62.3) | 583 (92.5) | 18941 (53.7) |
| **Rehydration method used in hospital** | | | | |
| Oral rehydration solution (ORS) | 885 (19.5) | 2062 (67.2) | 117 (18.6) | 25699 (72.9) |
| Intravenous (IV) fluid | 3655 (80.5) | 1008 (32.8) | 513 (81.4) | 9544 (27.1) |
| **Length of hospital stay** | | | | |
| <24 hours | 3111 (69.4) | 2432 (80.0) | 438 (70.1) | 26784 (79.1) |
| ≥24 hours | 1372 (30.6) | 607 (20.0) | 187 (29.9) | 7082 (20.9) |

ETEC cases has fluctuated, with a return towards 2008 rates, in 2022 (9.2%) in 2023 (7.5%). During the same time frame, the frequency of *V. cholerae* declined, reaching 4.5% in 2023 (**Fig 4**).

## The monthly distribution of ETEC, *V. cholerae,* and co-infection cases

**Fig 5** shows the distribution of ETEC, *V. cholerae,* and co-infection cases from January to December (2018–2023). Notably, there was a cholera outbreak in 2023. ETEC cases were also high in that period. During the peak months, the overall proportion of ETEC cases was 5% in April, 10% in May, and 24% in June. During the same study period, the proportion of patients from whom *V. cholerae* was identified was 4% in April, 6% in May, and 18% in June. The proportion of co-infection cases was 2% in April, 6% in May, and 25% in June.

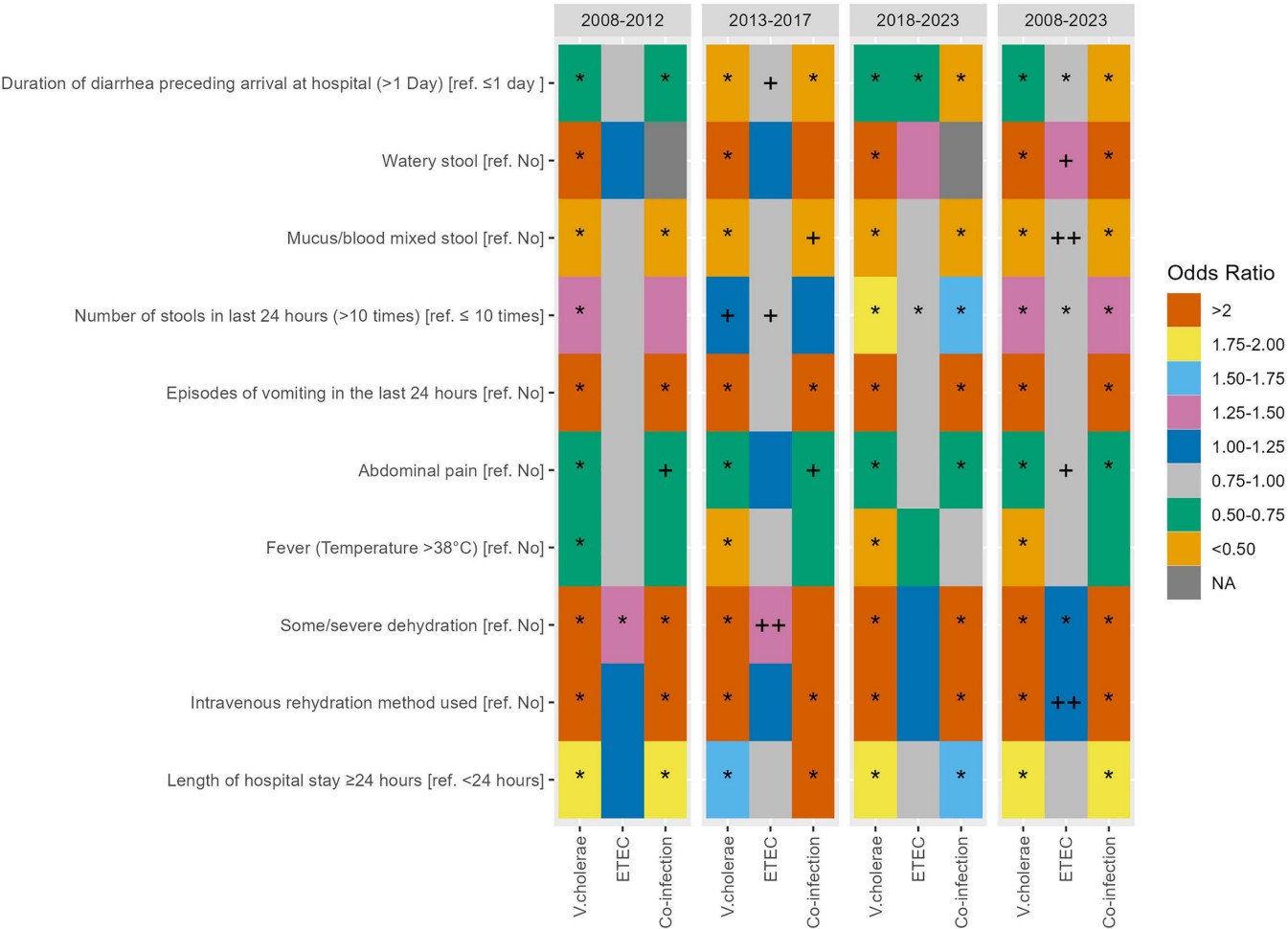

**Fig 3. Association of clinical outcomes of patients with diarrheal illnesses due to *V. cholerae*, ETEC and co-infection.** * *Indicates statistical significance at p < 0.001, ++ indicates statistical significance at p < 0.01, + indicates statistical significance at p < 0.05. * Adjusted for age , sex, area of residence, household asset index, use of rehydration solution before arrival, and use of antibiotics before hospitalization. In multiple logistic regression analysis, separate models were run for each outcome given in the first column. A total of thirty models (individual logistic regression) were performed for the final analysis. Outcome variables: Each clinical characteristic is considered as a dependent variable: Duration of diarrhea preceding arrival at hospital [≤1 day (reference), >1 Day], watery stool [no (reference), yes], blood mixed stool [no (reference), yes], number of stools in last 24 hours [≤10 times (reference), >10 times], vomiting in last 24 hours [no (reference), yes], abdominal pain [no (reference), yes], fever [no (reference), yes], some or severe dehydration [no (reference), yes], intravenous rehydration method used [no (reference), yes], length of hospital stay [<24 hours (reference), ≥24 hours]. Exposure variables: pathogens [V. cholerae (ref= no V. cholerae), ETEC (ref= no ETEC), co-infection with both cholera and ETEC (ref=No V. cholerae or ETEC)].*

## Discussion

Worldwide, acute watery diarrhea is often caused by various enteric bacterial pathogens, including ETEC, and *V. cholerae*, particularly in regions with poor sanitation and limited access to clean water [23,24]. Both organisms are typically studied separately. However, the phenomenon of simultaneous co-infection with both *V. cholerae* and ETEC is worthy of dedicated study because these patients are at increased risk for severe diarrheal symptoms [25,26]. Our analysis revealed that, among diarrheal patients with co-infections, stool frequency, vomiting, dehydration, and the requirement for intravenous rehydration were increased among co-infected patients.

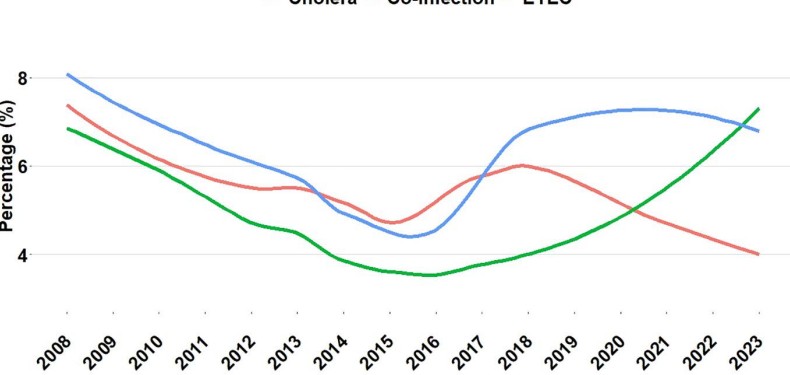

**Fig 4. Frequency (%) of patients infected with ETEC, *V. cholerae* and co-infection.**

Relative to patients without ETEC or *V. cholerae* we observed that reports of watery stools were four times greater when *V. cholerae* was the only infection, but eleven times higher when *V. cholerae* and ETEC patients were co-infected simultaneously. Literature suggests that co-infection with ETEC and *V. cholerae* may act synergistically, enhancing each organism's pathogenicity, thereby worsening clinical symptoms [27–29]. Mechanistically, the impact of co-infection with both ETEC and *V. cholerae* has been observed to result in more severe gastrointestinal inflammation while increasing risk for infection [30]. Overall, a combination of pathogenic and immunological factors is likely to account for the higher occurrence of watery stools in co-infections, relative to infections caused by ETEC or *V. cholerae* alone [10]. Findings from our study also revealed that clinical features associated with acuity, including the number of stools, vomiting, and the requirement of intravenous rehydration, were significantly increased among identified cases of co-infection.

The occurrence of watery stools was eleven times higher in cases of co-infection compared to infection with single pathogen. However, we also observed that the odds of dehydrating diarrhea were higher in cases caused by *V. cholerae* alone. This observation corresponds with prior reports in Bangladesh indicating that dehydrating diarrhea and intravenous fluid requirements are significantly associated with *V. cholerae* infection, among children aged 12–23 months [14]. Findings from a cohort profile among cholera patients in Yemen showed similar results [31]. Nearly all of these patients required hospitalization for more than 48 hours, similar to our study's observations [31].

Our current analysis also revealed age-specific differences between patients with *V. cholerae*, ETEC, or co-infection with both. We observed that the rate of co-infection among patients aged >15 years has increased in recent years between 2018–2023. Adults' more frequent exposure to contaminated environments outside of their homes may be the cause of this [32,33]. Individuals living in densely populated or resource-limited settings, where sanitation and hygiene practices may also be suboptimal, may experience increased environmental exposure to pathogens [34]. Additional factors not studied in this survey may also influence the acquisition of infections. These include variations in pathogen strains and host immune response [35].

A comparison of demographic characteristics revealed that majority of *V. cholerae*-infected diarrhea cases occurred in individuals aged 15 years and older. In contrast, 50% of ETEC cases occurred in children under five. ETEC diarrheal illnesses, which often goes underdiagnosed, are increased where WASH infrastructure is inadequate [13]. Our findings showed similar results. Studies have reported that in during seasonal peaks in Bangladesh adults requiring hospitalization more often present with cholera followed by ETEC diarrhea [36]. According to the hospital-based surveillance system, patients report co-infections too due to *V. cholerae* and ETEC [11].

The frequencies of *V. cholerae*, ETEC, and co-infection among hospitalized patients varied over time. On a yearly basis, the frequency of co-infection increased between 2017 and 2021, followed by a decline in 2023. Although

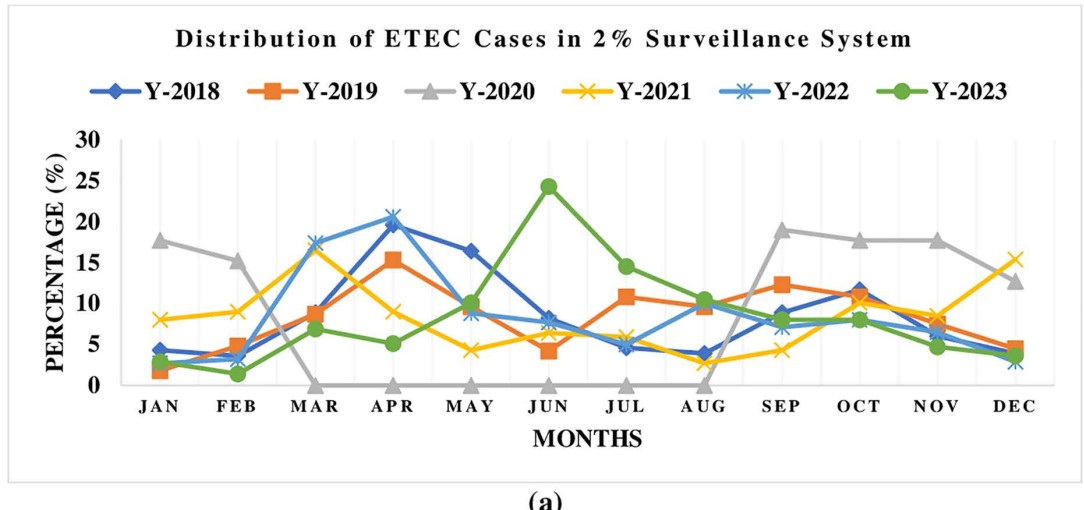

**(a)**

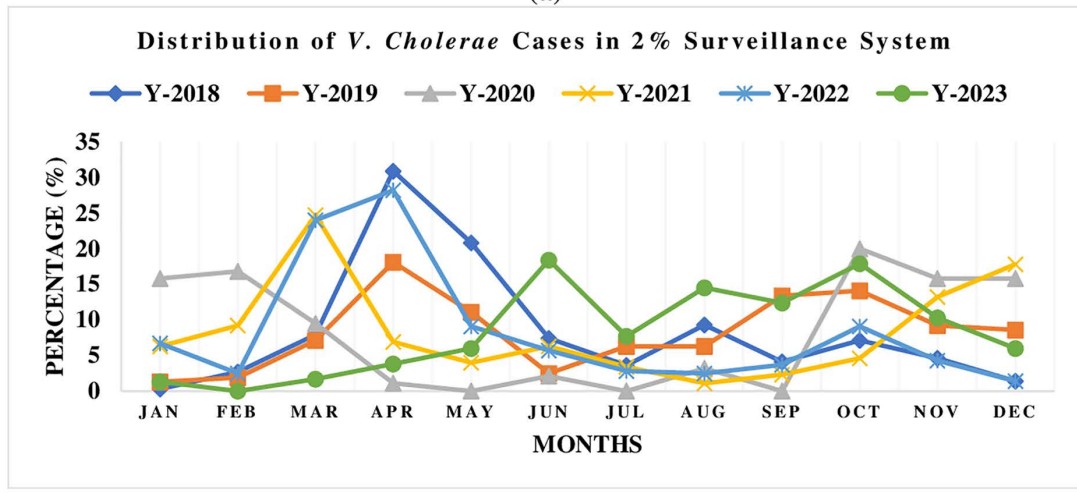

**(b)**

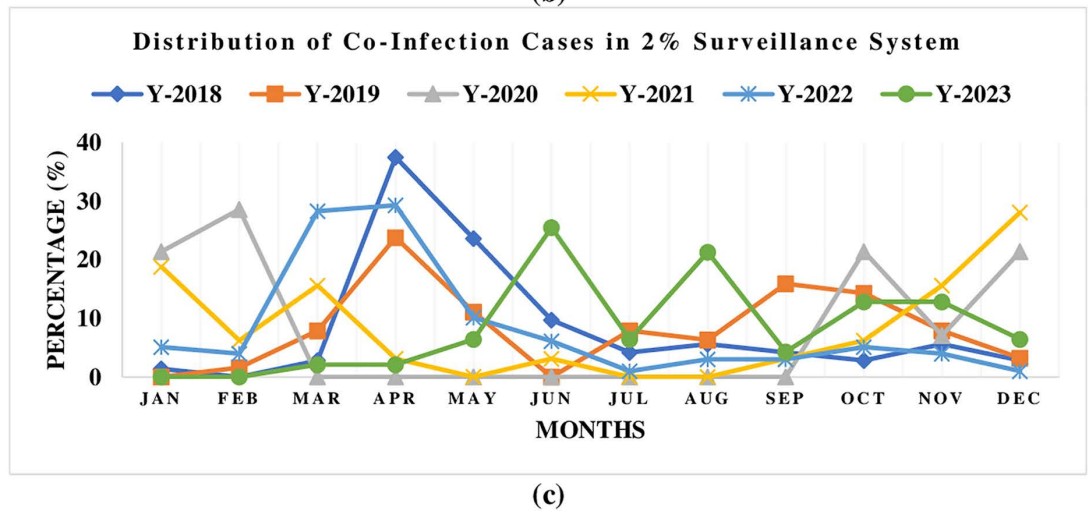

**(c)**

**Fig 5.  Distribution of ETEC (a),** *V. cholerae* **(b), and co-infection (c) cases in 2% surveillance system of the icddr,b Dhaka Hospital.**

research indicated a decline in rates of co-infection, our analysis of a large data set revealed that frequency of co-infection among admitted patients has been rising recently [14,37]. This study's findings also highlighted the monthly distribution of co-infection, ETEC infections alone, and *V. cholerae* infections. Our investigation found a higher proportion of hospitalized patients with pathogens of interest, including co-infections, from April to June, An outbreak may have contributed to the increased rate of co-infection observed in 2018 [38]. During extreme weather events, factors including flooding, poor drainage, and poor waste management can all lead to an increase in diarrheal diseases [39].

We observed that the duration of diarrhea before hospitalization was the shorter for co-infected patients compared to those with a single pathogen. The reason for this finding may be that interactions between the two pathogens (co-infection) lead to a more severe and rapid onset of symptoms relative to infections with a single pathogen [10]. This observation suggests that pathogenicity of ETEC and *V. cholerae* may be amplified during co-infection [10,11].

Strengths and limits: This study was undertaken using a robust dataset from the largest diarrheal disease hospital, spanning 16 years. While Amin et al.'s recent study analyzes ETEC infections and co-infections, including those with *V. cholerae*, using surveillance data from Dhaka from 2017 to 2022, our analysis covers a different time range (2008–2023) [40]. This approach allowed us to analyze trends, patterns, and changes over time. However, there are limitations to this data. Corresponding national statistics on disease severity were not available. Specifically, only patients with diarrhea who sought care at Dhaka Hospital were included in this analysis. We are mindful that observations made in this hospital-based population may diverge from incidence and prevalence patterns in the broader community. Likewise, there was limited information available regarding the follow-up of these patients. The absence of high-resolution climate data and detailed records of outbreak events or interventions limited our ability to assess their impact on year-to-year differences during the study period. The absence of quantitative data on pathogen load also limited our ability to distinguish between asymptomatic carriage and clinically significant infections. Information on the cholera vaccination status of patients was not available in the dataset, which restricts our ability to assess the potential impact of prior vaccination on susceptibility, disease severity, or pathogen distribution.

## Conclusion

Co-infection with *V. cholerae* and ETEC results in more severe clinical manifestations requiring intensive medical management compared to single-pathogen infections. These findings highlight the need for enhanced clinical preparedness and consideration of testing for both pathogens to optimize patient care. Our findings highlight the potential value of vaccines targeting ETEC and *V. cholerae* to improve protection.

## Supporting information

**S1 Table. Clinical features according to age group for pathogen.**
(DOCX)

**S2 Table. Association of clinical characteristics of patients with diarrheal illnesses due to *V. cholerae*, ETEC and co-infection (2008–2023).**
(DOCX)

## Acknowledgments

The authors would like to express their gratitude to all the participants, parents, caregivers, and families for their involvement and contributions to the study. We gratefully acknowledge icddr,b's core donors, the governments of Bangladesh and Canada, for providing core/unrestricted support and their ongoing commitment to icddr,b's public health research efforts.

## Author contributions

**Conceptualization:** Jinat Alam, Shamsun Nahar Shaima, Subhra Chakraborty, A. S. G. Faruque.

**Formal analysis:** Jinat Alam, Rina Das, Md. Ahshanul Haque, Md. Nasif Hossain, Soroar Hossain Khan.

**Investigation:** Jinat Alam, Shamsun Nahar Shaima.

**Methodology:** Jinat Alam, Shamsun Nahar Shaima, Rina Das.

**Software:** Jinat Alam, Rina Das, Md. Ahshanul Haque, Md. Nasif Hossain.

**Supervision:** Tahmeed Ahmed, Subhra Chakraborty, A. S. G. Faruque.

**Writing – original draft:** Jinat Alam, Shamsun Nahar Shaima, Rina Das.

**Writing – review & editing:** Sharika Nuzhat, Mohammod Jobayer Chisti, Tahmeed Ahmed, Subhra Chakraborty, A. S. G. Faruque.

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
