## [Decision Letter · Decision Letter 0]

16 Aug 2025

Epidemiology and clinical profile of diarrhea associated with enterotoxigenic Escherichia coli and Vibrio cholerae in Bangladesh: findings from a hospital-based surveillance system, 2008-2023

Dear Dr. Das,

Thank you for submitting your manuscript to PLOS Neglected Tropical Diseases. After careful consideration, we feel that it has merit but does not fully meet PLOS Neglected Tropical Diseases's publication criteria as it currently stands. Therefore, we invite you to submit a revised version of the manuscript that addresses the points raised during the review process.

Please submit your revised manuscript within 60 days Oct 15 2025 11:59PM. If you will need more time than this to complete your revisions, please reply to this message or contact the journal office at plosntds@plos.org. Please include the following items when submitting your revised manuscript:

We look forward to receiving your revised manuscript.

Kind regards,

Vangelis Economou

Guest Editor

Elsio Wunder Jr

Section Editor

Shaden Kamhawi

co-Editor-in-Chief

Paul Brindley

co-Editor-in-Chief

**Additional Editor Comments (if provided):**

**Journal Requirements:**

**Reviewers' Comments:**

Reviewer's Responses to Questions

**Key Review Criteria Required for Acceptance?**

**Methods**

-Are the objectives of the study clearly articulated with a clear testable hypothesis stated?

-Is the study design appropriate to address the stated objectives?

-Is the population clearly described and appropriate for the hypothesis being tested?

-Is the sample size sufficient to ensure adequate power to address the hypothesis being tested?

-Were correct statistical analysis used to support conclusions?

-Are there concerns about ethical or regulatory requirements being met?

Reviewer #1: Overall, the methods are well-explained and rely on a rigorously collected dataset at icddr,b. I do not have statistical expertise but the analysis seems reasonable.

- Why was the time period 2008-2023 chosen?

- Were multiple comparisons taken into account during the logistic regression analysis for setting the p-value threshold for statistical significance?

Reviewer #2: -Are the objectives of the study clearly articulated with a clear testable hypothesis stated? YES

-Is the study design appropriate to address the stated objectives? YES

-Is the population clearly described and appropriate for the hypothesis being tested? YES

-Is the sample size sufficient to ensure adequate power to address the hypothesis being tested? YES

-Were correct statistical analysis used to support conclusions? YES

-Are there concerns about ethical or regulatory requirements being met? YES

Reviewer #3: 1.The inclusion of parental education levels (both maternal and paternal) as covariates in regression models is not sufficiently justified in the manuscript. The theoretical or empirical relevance of these variables to disease occurrence, infection type, or clinical severity is unclear. In contrast, more clinically informative variables—such as duration of illness, presence of severe complications, ICU admission, or mortality—could provide a more comprehensive picture of disease burden and care needs.

2.The age stratification of children is rather coarse, primarily using 5 years as a cut-off. However, previous studies have shown that diarrheal etiology and clinical presentation can vary significantly across finer pediatric age groups (e.g., <6 months, 6–23 months, 24–59 months). It is recommended that the authors adopt a more granular age classification for children to improve the specificity and interpretability of the findings.

**Results**

-Does the analysis presented match the analysis plan?

-Are the results clearly and completely presented?

-Are the figures (Tables, Images) of sufficient quality for clarity?

Reviewer #1: The results presented are clear and organized in a logical manner.

Reviewer #2: -Does the analysis presented match the analysis plan? YES

-Are the results clearly and completely presented? YES

-Are the figures (Tables, Images) of sufficient quality for clarity? THEY ARE CLEAR WITH SOME EDITS SUGGESTED BELOW

Reviewer #3: Regarding the seasonal variation (Figure 5): the peak months for ETEC, V. cholerae, and co-infection appear inconsistent across years, with particularly notable spikes in 2023. The authors are encouraged to further elaborate on potential factors contributing to these year-to-year differences, such as abnormal weather patterns, outbreak events, changes in diagnostic protocols, or public health interventions.

**Conclusions**

-Are the conclusions supported by the data presented?

-Are the limitations of analysis clearly described?

-Do the authors discuss how these data can be helpful to advance our understanding of the topic under study?

-Is public health relevance addressed?

Reviewer #1: In general the conclusions are supported and the authors are careful not to overinterpret their data, which is ultimately correlative.

- An important limitation that was not discussed was the potential contribution of subclinical infections, especially for ETEC. Related is the point that pathogen load information is not available, further limiting interpretations.

- Is the study also limited by a lack of information on cholera vaccination status of patients?

- How surprising is the finding that V. cholerae associated symptoms tend to "dominate" when there is co-infection?

Reviewer #2: -Are the conclusions supported by the data presented? YES THEY ARE

-Are the limitations of analysis clearly described? LIMITATIONS HAVE BEEN CLEARLY OUTLINED AND discussed

-Do the authors discuss how these data can be helpful to advance our understanding of the topic under study? YES

-Is public health relevance addressed? YES

Reviewer #3: This study utilized diarrheal surveillance data from icddr,b hospital in Bangladesh spanning from 2008 to 2023, and analyzed the epidemiological patterns and clinical characteristics of diarrhea cases caused by enterotoxigenic Escherichia coli (ETEC) and Vibrio cholerae.

**Editorial and Data Presentation Modifications?**

Reviewer #1: - Please doublecheck figure references within the text for accuracy throughout.

- Line 253: "valuable insights" seems a little overblown, since the trend itself is not further interpreted.

- Please consider using a colorblind-friendly color scheme in the figures (especially Figure 3).

Reviewer #2: -To help readers understand the baseline characteristics, Table 1 provides good demographic characteristics, it would be beneficial to add a column for "Total Patients" or "No Pathogen Identified" to provide a complete picture of the study population from which these groups were drawn, especially since your logistic regression uses "no ETEC or V. cholerae" as a reference.

-The Denominator used should be clear for example, in the results section, "4540(11%) tested positive for V. cholerae, 3070(8%) had ETEC, and 630(1.5%) had co-infection"), it would be helpful to clarify what the denominator was used for percentage (i.e. 43,483 patients, or 8240 patients found to have one of the pathogens?) This is because abstract implies the former, but the results section states "Of these, 8240 were found to have a stool culture positive for ETEC, V. Cholerae, or both pathogens." Clarify this early on

-Consider rephasing the sentences, too much numbers and adds ratios can be difficult to read for example you can consider the line in the abstrati to read "Co-infected cases showed significantly higher odds of severe outcomes, including watery stools (aOR: 12.33), high stooling frequency (>10/day, aOR: 1.50), vomiting (aOR: 3.16), and intravenous fluid requirements (aOR: 8.70) compared to single-pathogen infections" making it easy to read

-Consistent Terminology: Ensure consistent capitalization and formatting for pathogen names (e.g., V. cholerae vs. V. Cholerae, ETEC). Standard scientific notation uses italics for genus and species. "Intravenous-fluid requirements" (abstract) vs. "intravenous rehydration" (results/discussion). Use one consistent term. "Hospital-stay" vs. "hospitalization" vs. "length of hospital stay."

Fig 2 and S1 Fig seem to have same information, I would suggest that if you keep them you at least say fig 2 is a summary and the detailed is S1 or just replace fig 2 with S1.

Reviewer #3: (No Response)

**Summary and General Comments**

Reviewer #1: In this manuscript, Alam and colleagues report a retrospective analysis of a large diarrheal disease dataset from icddr,b focused on clinical profiles and epidemiology of V. cholerae and ETEC co infections, which have been sporadically studied in the past but lacked large scale datasets.

The study is well-described and the results and conclusions appear robust. I have only minor comments for revision (listed in the different sections) apart from one major concern, which is that a very similar analysis of the same dataset from 2017-2022 was published earlier this year (https://www.sciencedirect.com/science/article/pii/S1201971224004405). This study included ETEC-V. cholerae coinfections as part of the analysis, finding that ETEC/V. cholerae co-infection was associated specifically with higher rates of vomiting and dehydration. This is consistent with what the current manuscript reports, which is a useful replication of the analysis, but this goes unaddressed in the submitted work. I do not think that this issue precludes publication, however, the authors must include a description of this study in their revised manuscript as well as a detailed discussion of how their study differs from the previously published work (e.g., timeframe, etc.) and whether the scope of the conclusions differ as well.

Reviewer #2: This is a great addition to the understanding of the to pathogens which recent literature has shown that in cholera outbreaks ETEC is also a key pathogen. These data and information from the analysis will go along way to argue for possible combination vaccines.

Reviewer #3: The current title implies that the study addresses the epidemiology and clinical profile of all diarrheal patients; however, the analysis is limited to those infected with ETEC and V. cholerae. Other common etiologic agents of diarrhea, such as rotavirus and norovirus, were neither addressed nor clarified as excluded. If the study focuses solely on ETEC and V. cholerae, this should be clearly reflected in the title, and the mention of the total number of diarrheal patients (43,483) should be presented with caution to avoid misleading readers.

PLOS authors have the option to publish the peer review history of their article (what does this mean? ). If published, this will include your full peer review and any attached files.

**Do you want your identity to be public for this peer review?** For information about this choice, including consent withdrawal, please see our Privacy Policy .

Reviewer #1: No

Reviewer #2: **Yes: ** Michelo Simuyandi

Reviewer #3: No

**Figure resubmission:**

**Reproducibility:**



---

## [Decision Letter · Decision Letter 1]

29 Oct 2025

Dear Dr. Das,

We are pleased to inform you that your manuscript 'Epidemiology and clinical profile of diarrhea associated with enterotoxigenic Escherichia coli and Vibrio cholerae in Bangladesh: findings from a hospital-based surveillance system, 2008-2023' has been provisionally accepted for publication in PLOS Neglected Tropical Diseases.

Best regards,

Elsio A Wunder Jr, DVM, Ph.D.

Section Editor

Elsio Wunder Jr

Section Editor

Shaden Kamhawi

co-Editor-in-Chief

Paul Brindley

co-Editor-in-Chief

Reviewer's Responses to Questions

**Key Review Criteria Required for Acceptance?**

**Methods**

-Are the objectives of the study clearly articulated with a clear testable hypothesis stated?

-Is the study design appropriate to address the stated objectives?

-Is the population clearly described and appropriate for the hypothesis being tested?

-Is the sample size sufficient to ensure adequate power to address the hypothesis being tested?

-Were correct statistical analysis used to support conclusions?

-Are there concerns about ethical or regulatory requirements being met?

Reviewer #1: (No Response)

**Results**

-Does the analysis presented match the analysis plan?

-Are the results clearly and completely presented?

-Are the figures (Tables, Images) of sufficient quality for clarity?

Reviewer #1: (No Response)

**Conclusions**

-Are the conclusions supported by the data presented?

-Are the limitations of analysis clearly described?

-Do the authors discuss how these data can be helpful to advance our understanding of the topic under study?

-Is public health relevance addressed?

Reviewer #1: (No Response)

**Editorial and Data Presentation Modifications?**

Reviewer #1: (No Response)

**Summary and General Comments**

Reviewer #1: The authors have responded fully to my review and I have no further comments. This is a useful addition to the literature and provides valuable clinical information on an understudied aspect of diarrheal disease.

PLOS authors have the option to publish the peer review history of their article (what does this mean? ). If published, this will include your full peer review and any attached files.

**Do you want your identity to be public for this peer review?** For information about this choice, including consent withdrawal, please see our Privacy Policy .

Reviewer #1: No

---

## [Editor Report · Acceptance letter]

Dear Dr. Das,

We are delighted to inform you that your manuscript, " 

Epidemiology and clinical profile of diarrhea associated with enterotoxigenic Escherichia coli and Vibrio cholerae in Bangladesh: findings from a hospital-based surveillance system, 2008-2023," has been formally accepted for publication in PLOS Neglected Tropical Diseases.

Best regards,

Shaden Kamhawi

co-Editor-in-Chief

Paul Brindley

co-Editor-in-Chief
